The chronic effects of change of direction during repeated-sprint training on jumping, sprinting, and change-of-direction abilities in players: a systematic review and meta-analysis

Kong Runzhou 1
Cao Lei 2
Li Dongyu ldyswim@163.com 1
1 Guangzhou Sport University , Guangzhou , China
2 Central South University, School of Humanities , Changsha , China
Gardasevic Jovan
Electronic publication date: 2025 May 20
Publication date: 2025
Volume: 13
Electronic Location ID: e19416
Received 2025 Jan 15; Accepted 2025 Apr 10
Copyright: ©2025 Kong et al.
Copyright year: 2025
Copyright holder: Kong et al.
License: This is an open access article distributed under the terms of the Creative Commons Attribution License, which permits unrestricted use, distribution, reproduction and adaptation in any medium and for any purpose provided that it is properly attributed. For attribution, the original author(s), title, publication source (PeerJ) and either DOI or URL of the article must be cited.
License URL: https://creativecommons.org/licenses/by/4.0/

Keywords: Multidirectional sprint protocols, Explosive movement measures, Neuromuscular adaptation

Funding: The authors received no funding for this work.

==============================
Objectives

This systematic review and meta-analysis aimed to compare the effects of repeated sprint training with one change of direction (RS-OCOD) and multidirectional changes of direction (RS-MCOD) on players’ jumping, sprinting, and change-of-direction (COD) ability.

Methods

Following PRISMA guidelines, 15 randomized controlled trials involving 223 players were analyzed. Standardized mean differences (SMD) and 95% confidence intervals (CI) were calculated using fixed- or random-effects models. Heterogeneity (I2 statistic), publication bias (funnel plots and Egger’s test), and sensitivity analyses were conducted. Subgroup analyses differentiated RS-OCOD and RS-MCOD outcomes.

Results

RS-COD training significantly improved countermovement jump (CMJ) (SMD = −0.68, 95% CI [−1.04 to −0.34]), 20–40 m sprinting ability (SMD = 0.70 [0.39, 1.01]), and COD ability (SMD = 0.77 [0.39, 1.16]). RS-MCOD demonstrated superior effects on 20–40 m sprinting ability (large effect: SMD = 0.97) and COD ability (large effect: SMD = 0.97), while RS-OCOD showed greater benefits for CMJ (large effect: SMD = −0.92). High heterogeneity (I2 > 70%) was observed in 5 m/10 m sprint analyses, attributed to methodological diversity and age variability. Prediction intervals indicated potential overestimation of RS-MCOD effects.

Conclusion

RS-OCOD can significantly enhance a player’s CMJ ability, while RS-MCOD can notably improve sprinting and COD abilities. These findings advocate for the incorporation of RS-COD into training programs to boost competitive performance. However, the effectiveness of RS-COD may vary based on the number and complexity of directional changes incorporated into the training regimen. Among them, RS-MCOD is prioritized for sports requiring frequent directional changes, whereas RS-OCOD enhances vertical jump performance in explosive sports.

Introduction

Repeated-sprint training (RST) can be utilized to prepare players for the intermittent, high-intensity demands of competition, which involve frequent accelerations, decelerations, and changes of direction (Taylor et al., 2017). RST has garnered increasing attention in recent scientific literature (Hoffmann Jr et al., 2014; Fernandez-Fernandez et al., 2012; Bishop, Girard & Mendez-Villanueva, 2011). It is operationally defined as a training modality involving ≥3 maximal-effort sprints (≤10 s duration) interspersed with brief (≤60 s) passive recovery period (Thurlow et al., 2023). Existing evidence indicates that RST acutely enhances the activity of key enzymes in both anaerobic and aerobic energy pathways (Burgomaster, Heigenhauser & Gibala, 2006; Gibala et al., 2006). Therefore, RST is a mixed training method that targets both neuromuscular and metabolic systems simultaneously (Thurlow et al., 2025), resulting in substantial improvements across a range of physical qualities that are important to sports performance, including speed, aerobic capacity, intermittent running performance, repeated-sprint ability, change of direction (COD) ability, and jump height (Thurlow et al., 2024).

In various ball sports (e.g., basketball, football, tennis, etc.), RST with COD (RS-COD) is essential due to the frequent accelerations and decelerations that involve COD (Taylor et al., 2015). It has been reported in the literature that elite soccer players change direction at angles less than 90° up to 300 times per game per side, and change direction at angles between 90° and 180° 45 to 49 times per game (Taylor et al., 2017). The acute demand of RS-COD depends on the number of direction changes, angles, distance between each direction change, and the duration of the sequence (Zagatto et al., 2017; Padulo et al., 2015; Attene et al., 2016). These factors affect the absolute speed gained and muscle work during sprinting, propelling, and braking. In addition, RS-COD involves the accumulation of acceleration and deceleration, which can increase neuromuscular demands (Buchheit, Haydar & Ahmaidi, 2012).

Thurlow et al. (2025) classified RST into three forms: straight line sprints, shuttle sprints and multidirectional sprints. And they suggested that shuttle sprints, by emphasizing changes of direction to limit absolute running speed, may improve COD ability to a greater extent than straight-line sprints, despite uncertain effects shown by the width of the confidence interval. Additionally, multidirectional sprint protocols can similarly enhance COD ability, acceleration, and deceleration, but the repeated distances should be limited to maintain the intensity of each effort (Thurlow et al., 2024).

Previous studies mainly compared RST with a control group to assess its training effects, or compared straight line sprints with RS-COD to explore the differences between RST with and without COD. Few have directly examined the impact of the number of direction changes in RST on physical qualities. Thus, this study aims to systematically review and meta-analyze the effects of RS-COD on jumping, sprinting, and COD abilities, and to compare the differences between one-direction (RS-OCOD) and multi-direction changes (RS-MCOD).

Information and Research Methods

Study selection

This meta-analysis followed the Cochrane Collaboration (Cumpston et al., 2019) and adhered to the PRISMA guidelines (Preferred Reporting Items for Systematic Reviews and Meta-Analyses) (Moher et al., 2010). The PICOS (Population, Intervention, Control, Outcome, Study Design) principles were used to define the inclusion and exclusion criteria, as shown in Table 1. The present systematic review and meta-analysis followed the PRISMA guidelines, and the protocol has been registered in PROSPERO (ID: CRD42024620508).

Search strategy

The systematic review and meta-analysis conducted searches in databases such as PubMed (Algorithm: Title/Abstract; Filters applied: Randomized Controlled Trail, English, Humans), Web of Science (Algorithm: Topic; Filters applied: Article or Dissertation Thesis, English, Sports Sciences), Scopus (Algorithm: TITLE-ABS-KEY, Filters applied: English, Chinese), ScienceDirect (Algorithm: Title/abstract/author-specified keywords; Filters applied: Research articles, English), and China National Knowledge Infrastructure (CNKI) (Algorithm: Topic, Filters applied: English, Chinese), SPORTDiscus (Algorithm: SU; Filters applied: English). The search time frame extended from the inception of records in each database to March 7, 2025, languages were limited to Chinese and English.

An example of the database search process was presented in Table 2. To ensure the accuracy of the search, two researchers cross-checked the search keywords. If there were any disagreements on keyword selection between the two researchers, a third researcher made the final decision. If necessary, manual searches were conducted to supplement the literature.

Table 1 Inclusion and exclusion criteria.

Project	Inclusion criteria	Exclusion criteria	
P	The pool of eligible participants included both male and female players who reported no existing injuries or illnesses and possessed a minimum competitive level corresponding to Tier 2 of the Participants Classification Framework (trained/developmental level)1, regardless of age.	We excluded players who were disabled, injured, or in an unhealthy state from consideration.
Furthermore, participants classified in tiers 0 or 1 of the Participants Classification Framework were also excluded.	
I	RST were interventions performed as standalone interventions, alongside normal training practice.
The intervention period was more than 2 weeks.	RST with no changes of direction was excluded.
Other interventions were included in the RST (e.g., RST + plyometric training).
The repetitive sprint training intervention consisted of both straight sprints and changes of direction sprints.	
C	The control group had no restrictions.	—	
O	The primary outcomes involve adaptations (medium to long term adaptations), concentrating on assessments of physiological or physical fitness levels at a minimum of two time points (baseline and post-intervention).
These adaptations include different aspects, including explosive power, sprint speed, aerobic capacity, anaerobic capacity, and the ability to change direction.	The results of acute adaptation, physiological indicators, and anatomical indicators were excluded	
S	Randomized parallel controlled trial	Self-controlled trial, correlation study	
Notes.

1 McKay et al. (2022).

Table 2 Web of Science literature selection strategy.

Program	Retrieval formula	
#1	TS = (“Repeated sprint training” OR “Straight Line Sprinting” OR “Straight Line” OR “repeated sprint” OR “Repeated sprint running” OR “repeated sprint” OR “intensive repeated sprint” OR “Change-of-Direction Sprinting” OR “Change-of-Direction” OR “Multi-Directional Sprint Training” OR “Multiple-sprint” OR “Multiple-sprint work” OR “Multiple-sprint running” OR “Repeated change of direction” OR “Shuttle-based sprints” OR “Repeated Shuttle Sprints” OR “shuttle run” OR “intermittent sprint” OR “Shuttle Sprints”)	
#2	TS = (player OR athletes OR “sportsman”)	
#3	TS = (“jump ability” OR “jump performance” OR “jump” OR “COD” OR “change-of-direction” OR “agility” OR “sprint” OR “sprint speed”)	
#4	#1 AND #2 AND #3	

Data extraction

Two researchers (R.K. and L.C.) searched each database using the same search formula and independently translated and reviewed the titles and abstracts for preliminary screening with Google Translate. After screening the literature in each database, the results were imported into EndNote X9 software for organization. Additionally, the researchers used Google Translate to review the full texts and determined which studies met the inclusion and exclusion criteria. The literature screening process was conducted by two investigators independently, and any disputes were subject to final decision by the third investigator (D.L.) based on the inclusion and exclusion criteria.

After confirming the final inclusion of all literature in the analysis, two researchers (R.K. and L.C.) extracted the data into a Microsoft Excel spreadsheet. The extracted data included the article title, year of publication, author(s) name, subject characteristics (e.g., age, number and type of participants, training protocol, period, outcome, results), training regimen (intervention period, training frequency, intervention), and pre- and post-testing data for outcome indicators (jumping ability, sprinting speed, and COD ability). The pre- and post-testing data were as mean ± standard deviation (mean ± SD). If the data were presented in pictures in the study, Getdata was used for data extraction. In instances where the data extracted by the two researchers (R.K. and L.C.) showed bias, a third researcher (D.L.) was assigned to review and finalize the data. When full-text articles or study data were unavailable, efforts were made to obtain the necessary content by contacting the authors via email.

Assessment of risk of bias

Two independent researchers (R.K. and L.C.) employed the Cochrane Risk of Bias Tool within Review Manager software version 5.4 to assess the methodological rigor of each eligible randomized controlled trial (RCT). The risk of bias across the included studies was evaluated based on seven criteria: (1) generation of random sequences, (2) concealment of allocation, (3) blinding of participants and staff, (4) blinding of outcome assessment, (5) incomplete outcome data, (6) selective reporting, and (7) any other sources of bias. The quality of the literature was determined by the number of studies classified as “low risk of bias” (with scores of ≥4 considered low risk, 2–3 as moderate risk, and 0–1 as high risk). These findings were crucial in establishing the overall quality assessment of the literature.

Statistical analysis

Data analysis was conducted using Review Manager software version 5.4, Stata 15.0 and R 4.4.3. We employed the standardized mean difference (SMD) along with 95% confidence intervals (CI) as the metric for aggregating effects on continuous outcome variables. The study conducted by Sanchez-Sanchez et al. (2019) included two RS-MCOD groups, which were combined in accordance with the recommendations outlined in the Cochrane Handbook (detailed formulas refer to Table 6.5.a of the Cochrane Handbook) (Chandler et al., 2019). Heterogeneity across studies was assessed using the I2 statistic, categorized as low (0% to 25%), moderate (25% to 75%), or high (75% to 100%) (Higgins et al., 2003). The fixed-effects model was applied for outcome indicators when I2 was less than 25%; otherwise, a random effects model was utilized. Statistical significance was set at p < 0.05. Effect sizes (ES) were classified based on Hedge’ g index into small (0.2 to 0.5), moderate (0.5 to 0.8), or large (above 0.8). Prediction intervals (PI) were computed alongside the estimates to convey the likely range of the true change in similar future studies. The Cochrane Risk of Bias Tool was employed to evaluate the methodological quality of the included studies, ensuring their internal and external validity. Between-study subgroup analysis to clarify the impact of RS-OCOD and RS-MCOD on final outcomes. Publication bias was assessed using funnel plots and Egger’s test. In cases where publication bias was detected, the trim-and-fill procedure by Duval and Tweedie was applied to adjust the funnel plot, with the bias reassessed after adjustment. A sensitivity analysis was performed to evaluate the stability and reliability of the pooled results from the meta-analysis.

Results

Literature search

A total of 2,983 studies were retrieved (Fig. 1). After exclusion and initial screening, 134 articles were left for further screening, and finally 27 possible articles were read in full. After full-text screening, four studies were correlation study (Padulo et al., 2015; Brini, Delextrat & Bouassida, 2021; Padulo et al., 2016; Dal Pupo et al., 2013) and three studies were self-control study (Born et al., 2016; Taylor et al., 2016; Ozgunen et al., 2021), therefore excluded. Two studies were excluded because the subjects were Tier 1 participants (Chen, 2019; Shi, 2021). Five studies were excluded because they did not meet the inclusion criteria for the intervention protocol. One study involved the RST without a COD (Tian, 2020), while the other four studies involved the RST in conjunction with other interventions (Brini et al., 2022; Brini et al., 2023; Morais et al., 2025; Kilit et al., 2025). The details of the excluded literature are provided in the Annex 1. Finally, 15 studies (Attene et al., 2016; Sanchez-Sanchez et al., 2019; Arede et al., 2021; Attene et al., 2015; Beato et al., 2019a; Beato et al., 2019b; Brini et al., 2020; Negra et al., 2022; Pavillon et al., 2021; Sagelv et al., 2019; Stanković et al., 2024; Zhou et al., 2024; Chtara et al., 2017; Young et al., 2013; Zhao, 2022), meeting the eligibility criteria, were included for further analysis.

Figure 1 PRISMA flow chart for inclusion and exclusion of studies.

Study characteristics

A total of 15 studies involving 223 players were ultimately included in this article, as they met the inclusion criteria for the analysis. The detailed inclusion and exclusion process is illustrated in Fig. 1 below. The RS-OCOD group (N = 124) received repeat sprint training with one change of direction, while the RS-MCOD group (N = 99) received repeat sprint training with multiple change of direction (Table 3). Furthermore, all included studies were randomized parallel controlled trials. Soccer was the most frequently studied sport, comprising 10 studies (66.7%), followed by basketball with four studies (26.7%) and tennis with 1 study (6.7%). Nine studies (60%) focused on trained or developmental players, while three studies (20%) involved highly trained or national-level players, and another three studies (20%) included elite or international-level players. Five studies (33.3%) examined adult players, whereas 10 studies (66.7%) focused on adolescent players. Only three studies (20%) included female players.

Table 3 Summary of reviewed studies including study design, level of evidence, period, outcome and result.

Study	Participants	Age (yrs)	Intervention protocol	Week/ fre	Outcomes	Test equipment	
			RS-OCOD	RS-MCOD				
Arede et al. (2021)	Amateur senior female basketball players	<19	Two sets of 10*10+10 m (one COD of 180°) with 30 s and 3 min recovery between sprints and sets	——	4/3	Jumping ability:
CMJ (cm)	Infrared optical system (OptoJump Next-Microgate, Bolzano, Italy)	
Sprint speed:
T10 (s), T25 (s)	Photoelectric cells (Witty, Microgate, Bolzano, Italy)	
Agility:
COD180R (s), COD180L (s)	
Attene et al. (2015)	Young male basketball players	16 ± 1	Three sets of 6-8*15+15 m (one COD) with 20 s and 4 min recovery between sprints and sets	Three sets of 6-8*10+10+10 m with 30 s of passive recovery	4/2	Jumping ability:
CMJ (cm), SJ (cm)	Accelerometer Myotest™ (Myotest SA, Sion, Switzerland)	
Attene et al. (2016)	Young male (n = 14) and female (n = 22) basketball players	16.1 ± 0.9	Three sets of 6-8*15+15 m (one COD) with 20 s and 4 min recovery between sprints and sets	Three sets of 6-8*30 m (6*5m) with 30 s of passive recovery	4/2	Jumping ability:
CMJ (cm), SJ (cm)	Accelerometer Myotest™ (Myotest SA, Sion, Switzerland)	
Beato et al. (2019a)	Elite level young soccer players	18–21	Three sets of 7*20+20 m (one COD of 180°) with 20 s and 4 min recovery between sprints and sets	——	8/1	Sprint speed:
T10 (s), T30 (s), T40 (s)
Agility:
505COD (s)	Infrared timing gates (Microgate, Bolzano, Italy)	
Beato et al. (2019b)	Amateur soccer players	21 ± 2.4	Three sets of 7*20+20 m (one COD of 180°) with 20 s and 4 min recovery between sprints and sets	——	2/2	Sprint speed:
T10 (s), T20 (s)
Agility:
505COD (s)	Infrared timing gates (Microgate, Bolzano, Italy)	
Brini et al. (2020)	Professional male basketball players	22.06 ± 2.8	——	Three sets of 8*30 m (five COD of 90°) with 20 s and 4 min recovery between sprints and sets	12/2	Jumping ability:
CMJ (cm), SJ (cm)	Optoelectric system (Opto-Jump Microgate, Italy)	
Agility:
T-test (s)	Electronic timing system (Brower Timing Systems, Salt Lake City, UT, USA)	
Sanchez-Sanchez et al. (2019)	Youth soccer players	14.7 ± 0.5
14.4 ± 0.5	——	Three sets of 10*18 m (two COD of 90°) with 18 m low-intensity recovery running and 4 min passive recovery between sprints and sets	8/2	Agility:
COD test (s)	Timing gates (Sport Test software, version 3.2.1; DSD
Laser System)	
Negra et al. (2022)	Youth male soccer players	15.6 ± 0.3	Two sets of 10*10+10 m with 5 s and 25 s recovery between sprints and sets	——	9/2	Sprint speed:
T10 (s), T20 (s)
Agility
505COD	Electronic timing system (Microgate srl, Bolzano, Italy)	
Pavillon et al. (2021)	Elite young male soccer players	One:
15.5 ± 2.7
Multi:
15.9 ± 1.7	Two sets of 10*10+10 m with 5 s and 25 s recovery between sprints and sets	Four sets of 10*20 m (with 3 directions) with 5 s and 25 s recovery between sprints and sets	30/6	Jumping ability:
CMJ (cm)	Smart jump portable force platform (Fusion Smart Speed Jump Fusion Sport, Australia)	
Sprint speed:
T5 (s), T10 (s)
Agility:
SPS (s)	Photoelectric cells (Smart Speed, Fusion Sport, Australia)	
Sagelv et al. (2019)	High-level male junior football players	17.4 ± 0.7	——	Four sets of 10*20 m (with 3 directions) with 5 s and 25 s recovery between sprints and sets	22/2	Sprint speed:
T10 (s), T20 (s)	Photocells (TC-Timer, Bower Timing System, Draper, UT, USA)	
Stanković et al. (2024)	Highest level female soccer players	19.2 ± 4.2	——	Three sets of 4*5+5+20 m with 30 s and 5 s recovery between sprints and sets	6/2	Jumping ability:
CMJ (cm), SJ (cm)	Two photoelectric cells (Optojump, Microgate, Bol”-zano, Italy)	
Sprint speed:
T10 (s), T20 (s), T30 (s)
Agility:
Pro-agility test (s), Zig-zag test (s), 9-6-3-6-9 sprint (s)	Photocell infrared timing gates (Microgate, Polifemo Radio Light, Bolzano, Italy)	
Zhou et al. (2024)	Collegiate tennis players	21.9 ± 1.66	——	15, 20 or 25 s linear running with three COD of 180° turn during each interval running	3/3	Sprint speed:
T5 (s)
Agility:
T-drill test (s), Spider run test (s), Termed reactive agility (s)	Smart Speeds	
Chtara et al. (2017)	Male elite-level field soccer players	13.6 ± 0.3	Two to four sets of 5-6*20-30 m (180° turns) sprints interspersed with 20s of passive recovery	——	6/2	Sprint speed:
T10 (s), T30 (s)	Photocell gates (Brower Timing Systems, Salt Lake City, Utah, USA	
Agility:
Zig-zag test (s)	Electronic timing gates (Brower Timing System, Salt Lake City, UT)	
Zhao (2022)	Young football players	U17	Three sets of 5-7*30-40 m (180° turns) sprints interspersed with 20-25s of passive recovery and 4 min recovery between sprints and sets		8/2	Jumping ability:
CMJ (cm), SJ (cm)	Smart jump portable force platform (Fusion Smart Speed Jump Fusion Sport, Australia)	
	Sprint speed:
T5 (s), T10 (s), T30 (s)	Electronic timing gates (Brower Timing System, Salt Lake City, UT)	
Young et al. (2013)	Well-trained soccer player	21.2 ± 2.6		Two sets of 4-8*40 m (with 4 directions) with 2 min and 10 min recovery between sprints and sets	8/1	Jumping ability:
CMJ (cm)	AMTI force platform; model OR6-5-1 (Watertown, USA)	
	Sprint speed:
T20 (s)	Infrared photocells	
Notes.

CMJ countermovement jump

SPS Slalom sprint test

R right

L left

T5/T10/T20/T25/T30/T40, 5-/10-/20-/25-/30-/40-meter sprint.

Risk of bias in the included articles

The Cochrane Risk of Bias assessment tool was utilized to evaluate the quality of the literature included in this meta-analysis. All studies considered in this analysis were randomized controlled trials that clearly described their methods for allocation concealment. Among these studies, only one reported the use of blinding for both researchers and participants. Regarding attrition bias, seven studies were found to be free of such bias, while one study demonstrated a high level of attrition bias. Within the scope of this meta-analysis, the body of included studies consisted of nine studies identified as having a low risk of bias and five studies with a moderate risk of bias, as illustrated in Fig. 2.

Meta-analysis results

The impact of RS-COD training on players’ jumping ability

A cumulative total of eight studies, comprising 15 experimental groups and involving 250 participants, were included in this meta-analysis to evaluate the impact of RS-COD on players’ jumping ability, as illustrated in Fig. 3. The aggregated effect sizes indicated that RS-COD had a moderate effect on countermovement jump (CMJ) (SMD = −0.69, 95% CI [−1.04 to −0.34]; I2 = 51.7%, p = 0.01). RS-OCOD demonstrated a large effect on CMJ (SMD = −0.92, 95% CI [−1.52 to −0.32]; I2 = 61.5%, p = 0.02), while RS-MCOD exhibited a moderate effect on CMJ (SMD = −0.53, 95% CI [−0.95 to −0.10]; I2 = 44.9%, p = 0.08). The aggregated effect sizes indicated that RS-COD had a small effect on SJ (SMD = −0.27, 95% CI [−0.56–0.02]; I2 = 0%, p = 0.51). Both RS-OCOD (SMD = −0.22, 95% CI [−065–0.21]; I2 = 0%, p = 0.98) and RS-MCOD (SMD = −0.29, 95% CI [−0.81–0.24]; I2 = 41.8%, p = 0.16) demonstrated a small effect on SJ. The prediction interval results were consistent with those of the meta-analysis, suggesting that the effect sizes of future studies on CMJ and SJ may not be statistically significant. Additionally, the prediction intervals for the subgroups indicated a consistent outcome.

Figure 2 Risk of bias assessment chart.

Note: Arede et al. (2021); Attene et al. (2015); Attene et al. (2016); Beato et al. (2019a); Beato et al. (2019b); Brini et al. (2020); Chtara et al. (2017); Sanchez-Sanchez et al. (2019); Negra et al. (2022); Pavillon et al. (2021); Sagelv et al. (2019); Stanković et al. (2024); Young et al. (2013); Zhao (2022); Zhou et al. (2024).

Figure 3 Forest plot of the effect of RS-COD on players’ jumping ability.

Note: Arede et al. (2021); Attene et al. (2015); Attene et al. (2016); Brini et al. (2020); Chtara et al. (2017); Sanchez-Sanchez et al. (2019); Pavillon et al. (2021); Stanković et al. (2024); Young et al. (2013); Zhao (2022).

The impact of RS-COD training on players’ sprinting ability

A total of eleven studies, which included 28 experimental groups and 347 participants, were evaluated to determine the impact of RS-COD on players’ sprinting ability, as illustrated in Fig. 4. The aggregated effect sizes indicated that RS-COD had a moderate effect on 5 m sprint speed (SMD = 0.69, 95% CI [−0.50–2.75]; I2 = 84.8%, p = 0.00). RS-OCOD demonstrated had a small effect on 5 m sprint speed (SMD = −0.31, 95% CI [−2.21–1.59]; I2 = 87.6%, p = 0.00), while RS-MCOD exhibited a large effect (SMD = 1.62, 95% CI [0.50–2.75]; I2 = 71.6%, p = 0.01). The aggregated effect sizes indicated that RS-COD had a small effect on 10 m sprint speed (SMD = 0.34, 95% CI [−0.11–0.78]; I2 = 67.8%, p = 0.00). Both RS-OCOD (SMD = 0.29, 95% CI [−0.24–0.82]; I2 = 65.1%, p = 0.00) and RS-MCOD (SMD = 0.42, 95% CI [−0.48–1.32]; I2 = 76.5%, p = 0.00) demonstrated a small effect on 10 m sprint speed. The aggregated effect sizes indicated that RS-COD had a moderate effect on 20–40 m sprint speed (SMD = 0.70, 95% CI [0.39–1.01]; I2 = 26.5%, p = 0.19). RS-OCOD demonstrated a moderate effect on 20–40 m sprint speed (SMD = 0.53, 95% CI [0.21–0.86]; I2 = 0%, p = 0.52), while RS-MCOD exhibited a large effect (SMD = 0.97, 95% CI [0.37–1.58]; I2 = 46.2%, p = 0.13). The prediction interval results were consistent with the meta-analysis results, which suggests that the effect size of future studies may not be statistically significant. The prediction interval results were consistent with those of the meta-analysis, suggesting that the effect sizes of future studies on 5 m, 10 m, and 20–40 m sprints may not be statistically significant. Additionally, the results of the 20–40 m prediction interval yielded varying outcomes, and the prediction interval for the RS-OCOD subgroup confirmed its significantly enhanced effect (Supplementary file S2).

Figure 4 Forest plot of the effect of RS-COD on players’ sprinting ability.

Note: Arede et al. (2021); Attene et al. (2015); Attene et al. (2016); Beato et al. (2019a); Beato et al. (2019b); Brini et al. (2020); Chtara et al. (2017); Sanchez-Sanchez et al. (2019); Negra et al. (2022); Pavillon et al. (2021); Sagelv et al. (2019); Stanković et al. (2024); Young et al. (2013); Zhao (2022); Zhou et al. (2024).

The impact of RS-COD training on COD ability

A total of ten studies, comprising 19 experimental groups and involving 259 participants. It was conducted to evaluate the impact of RS-COD on players’ COD ability, as illustrated in Fig. 5. The aggregated effect sizes indicated that RS-COD had a moderate effect on COD ability (SMD = 0.77, 95% CI [0.39–1.16]; I2 = 71.7%, p = 0.00). RS-OCOD demonstrated a small effect on agility (SMD = 0.54, 95% CI [−0.15–1.22]; I2 = 77.5%, p = 0.00), while RS-MCOD exhibited a large effect (SMD = 0.97, 95% CI [0.56–1.38]; I2 = 60.2%, p = 0.00). The prediction interval results were consistent with the meta-analysis results, which suggests that the effect size of future studies may not be statistically significant. The prediction interval results were consistent with those of the meta-analysis, suggesting that the effect sizes of future studies may not be statistically significant. Additionally, the prediction intervals for the subgroups indicated a consistent outcome.

Figure 5 Forest plot of the effect of RS-COD on players’ COD ability.

Note: Arede et al. (2021); Attene et al. (2015); Attene et al. (2016); Beato et al. (2019a); Beato et al. (2019b); Brini et al. (2020); Chtara et al. (2017); Sanchez-Sanchez et al. (2019); Negra et al. (2022); Pavillon et al. (2021); Sagelv et al. (2019); Stanković et al. (2024); Young et al. (2013); Zhao (2022); Zhou et al. (2024).

Reporting bias

Upon visual inspection of the funnel plot (Fig. 6), indications of publication bias were observed. The distribution of effect sizes for explosive force was asymmetrical, with the majority of effect sizes skewed to the left side of the funnel, resulting in an irregular shape of the funnel plot. Egger’s test confirmed these visual observations, yielding significant results for jumping ability (p = 0.04). In contrast, no significant publication bias was detected for sprinting ability (p = 0.46) or COD ability (p = 0.06). The presence of publication bias, particularly concerning jumping ability, was further supported by the funnel plot analysis. Notably, after four iterations using the linear method, the software estimated that there were no missing studies, resulting in a count of zero. Following the pruning and filling procedure, no adjustments were made to the funnel plot, and the combined results remained unchanged. This suggests that the aggregated outcomes of the meta-analysis are robust and reliable.

Figure 6 Funnel plot of the effect of RS-COD on players’ (A) jumping, (B) sprinting, and (C) COD abilities.

Sensitivity analysis

A sensitivity analysis was conducted using Stata 15.0 to assess the impact of each individual study on the outcomes of the meta-analysis, as illustrated in Fig. 7. The analysis revealed that the sensitivity of RS-COD in relation to players’ jumping ability is approximately −0.49, indicating a stable relationship within the study. Similarly, the sensitivity of RS-COD concerning players’ sprinting ability was found to be around 0.58, while the sensitivity for COD ability was approximately 0.77. These sensitivity values highlight the relative stability and reliability of the data included in this meta-analysis. The results of this sensitivity analysis strengthen confidence in the robustness of the aggregated findings presented.

Figure 7 Sensitivity analysis of the effect of RS-COD on players’ (A) jumping, (B) sprinting, and (C) COD abilities.

Note: Arede et al. (2021); Attene et al. (2015); Attene et al. (2016); Beato et al. (2019a); Beato et al. (2019b); Brini et al. (2020); Chtara et al. (2017); Sanchez-Sanchez et al. (2019); Negra et al. (2022); Pavillon et al. (2021); Sagelv et al. (2019); Stanković et al. (2024); Young et al. (2013); Zhao (2022); Zhou et al. (2024).

Discussion

This systematic review and meta-analysis is the first to compare the effects of RS-OCOD and RS-MCOD on players’ jumping, sprinting, and COD abilities. The analysis included 15 studies involving 223 players. The overall effect estimates indicated that RS-COD effectively improved CMJ, 20–40 m sprinting ability, and COD ability. However, since none of these results were confirmed by prediction intervals, the interpretation of these meta-analysis findings should be approached with caution and an objective perspective. This also underscores the necessity for further research to better estimate the differential effects of RS-OCOD and RS-MCOD. The subgroup analyses in this study were pre-planned to prevent data-driven exploration and minimize Type I errors. Our results indicated that both RS-OCOD and RS-MCOD significantly improved CMJ, 20–40 m sprinting, and COD abilities, with RS-MCOD also showing a notable enhancement in 5 m sprinting ability. The prediction interval for RS-OCOD in 20–40 m sprinting ability suggests that its effects may persist in future studies. A comparison of effect sizes revealed that RS-OCOD was more effective than RS-MCOD in enhancing CMJ, while the reverse was true for 20–40 m sprinting ability and COD ability. Based on these findings, we recommend prioritizing RS-OCOD for sports that require explosive jumping, such as basketball and volleyball. Conversely, for sports that involve frequent directional changes, such as football and basketball, RS-MCOD is the preferred training method.

Jumping ability

This meta-analysis evaluated the effects of RS-COD on players’ jumping ability. The results demonstrated that both RS-OCOD and RS-MCOD improved CMJ height, with RS-OCOD exhibiting a superior enhancement compared to RS-MCOD. The analysis revealed potential publication bias in the jumping ability data, which was adjusted using the trim-and-fill method. No additional studies were imputed during this adjustment, and sensitivity analysis confirmed the robustness of the findings, indicating relatively stable outcomes for meta-analysis of jumping ability. We conducted a self-assessment to identify potential causes of publication bias. First, the inclusion of both CMJ and SJ metrics in the jumping ability analysis may have contributed to heterogeneity. Second, duplicate inclusion of three studies was identified: two studies (Attene et al., 2016; Attene et al., 2015) were repeatedly incorporated due to reporting dual jump metrics, and one study (Pavillon et al., 2021) was redundantly included for analyzing three distinct population subgroups. These factors likely explain the observed publication bias in the jumping ability outcomes.

Previous studies suggest that most players benefit from increased exposure to varied movement combinations and repetitive high-intensity neuromuscular activation, which leads to short-term improvements in jumping ability (Arede et al., 2021). Specifically, enhancements in neural drive are associated with greater recruitment of agonist muscles, elevated neuronal firing rates, and improved synchronization of neural firing timing (Haff, Triplett & NSCA, 2016). Similarly, central adaptations induced by heightened neural activation can increase motor unit activation, resulting in enhanced muscle contraction intensity and stiffness (Haff, Triplett & NSCA, 2016). Attene et al. (2015) directly compared RS-OCOD and RS-MCOD (involving two 180° turns) and found that both significantly improved players’ CMJ, while only RS-MCOD enhanced SJ height. Furthermore, RS-MCOD demonstrated a notable advantage over RS-OCOD in CMJ improvement. In a follow-up study with increased directional complexity (six directions), the researchers observed comparable effects of RS-OCOD and RS-MCOD on jumping ability (Attene et al., 2016). Pavillon et al. (2021) reported similar conclusions in their investigation of RS-COD with three directional changes.

In the present study, the effect size comparison indicated that RS-OCOD may outperform RS-MCOD in CMJ improvement, although this difference was not confirmed by prediction intervals. CMJ is inherently a single explosive action requiring precise coordination between the braking (deceleration) and concentric (acceleration) phases. RS-OCOD may maximize the utilization of the stretch-shortening cycle, enabling muscles to generate higher power output within shorter timeframes, which plausibly explains its superior efficacy in CMJ enhancement observed in this analysis.

Sprinting ability

This meta-analysis evaluated the effects of RS-COD on players’ sprinting ability. The results demonstrated that both RS-OCOD and RS-MCOD improved 20–40 m sprint speed in players, with effect size magnitudes indicating superior efficacy of RS-MCOD over RS-OCOD. Furthermore, RS-MCOD exhibited additional benefits in 5 m sprint speed.

The observed improvements in ≥20 m sprint speed across both protocols may be attributed to cumulative adaptations from RST. Previous studies have established that repeated sprint efforts induce increases in muscle metabolites (e.g., phosphocreatine and glycogen* 32%) and enzymatic activity (Rodas et al., 2000). The total single-session sprint distances (20–40 m) employed in the included RS-COD interventions align closely with the outcome measures demonstrating significant effects, which may explain the primary training-induced enhancements in 20–40 m sprints. The superior effect sizes observed in RS-MCOD compared to RS-OCOD could be associated with the higher frequency of COD components in RS-MCOD protocols. Compared to mechanical straight-line repetitions (Henz & Schöllhorn, 2016) multidirectional movement patterns (e.g., direction-altered sprints) elicit greater neuromuscular (Horst et al., 2016) and neurophysiological adaptations through enhanced storage of elastic energy during eccentric phases, thereby facilitating greater kinetic energy release during concentric phases. However, prediction intervals suggest potential overestimation of RS-MCOD effects while confirming RS-OCOD efficacy. These mechanisms may also account for the additional 5 m sprint speed improvements observed with RS-MCOD. Notably, the high methodological repetitiveness among studies investigating 5 m sprint speed in this meta-analysis (as reflected in prediction intervals indicating potential null effects in future studies) limits definitive conclusions, suggesting that RS-MCOD may only potentially induce positive effects on 5 m sprint speed. Therefore, further experimental investigations are warranted to clarify RS-COD’s impacts on 5 m sprint speed.

Substantial heterogeneity was observed in meta-analyses of 5 m and 10 m sprint speeds. This may partially stem from our a priori decision to stratify interventions as subgroups, which inherently introduces methodological heterogeneity and justified our selection of random-effects models. Additionally, while we specifically selected player populations, the broad age range of included participants (13–22 years in 10 m sprint studies) likely contributed to the observed heterogeneity.

Change of direction ability

This meta-analysis evaluated the effects of RS-COD on players’ COD ability. The results demonstrated that RS-COD effectively enhances players’ COD ability, with RS-MCOD emerging as the primary influential modality for COD improvement. COD ability constitutes a critical physical competency component in numerous sports (Brughelli et al., 2008). RS-COD can be incorporated into RST programs to emphasize directional changes, limit absolute running velocity, and induce physiological demands comparable to RS-SS (Thurlow et al., 2023). Post-RST enhancements in explosive physical capacities may originate from neuromuscular and morphological adaptations. Neural adaptations may involve increased motor unit recruitment, firing frequency, and synchronization (Schmidt, 2004), while morphological changes could include shifts toward type IIa muscle fibers and increased muscle cross-sectional area (Ross & Leveritt, 2001). These mechanisms may explain RS-COD’s superior efficacy in improving players’ COD capabilities. Notably, the acute demands of RS-COD execution depend on the number and angles of directional changes, inter-change distances, and sequence duration (Padulo et al., 2015; Attene et al., 2016; Buchheit et al., 2010), all of which influence absolute velocities and muscular work during acceleration, propulsion, and deceleration phases. The integration of COD into RST accumulates acceleration-deceleration cycles, potentially amplifying neuromuscular demands (Buchheit, Haydar & Ahmaidi, 2012).

The effect size magnitudes indicated RS-MCOD as an effective approach for enhancing players’ COD ability, whereas RS-OCOD demonstrated limited efficacy. During training interventions, both the frequency of COD maneuvers and turning angles serve as deterministic factors for muscular adaptation (Stanković et al., 2022). RS-MCOD’s requirement for frequent directional changes promotes superior reactive strength development, optimizes pre-activation mechanisms, and enhances joint stiffness (Arede et al., 2021). Furthermore, RS-MCOD protocols involving multiple directional alterations (e.g., 3–5 consecutive turns of 90°–180°) better replicate real-game scenarios, comprehensively stimulating players’ deceleration-acceleration capacity, center-of-mass control, and dynamic balance, thereby significantly improving COD ability. However, substantial heterogeneity observed in forest plot interpretations introduced potential analytical errors. We attribute this heterogeneity primarily to methodological variations in COD assessment tools (Table 2). The included studies utilized diverse COD evaluation metrics: COD test, T-test, spider run test, zig-zag test, pro-test, 9-6-3-9 test, and slalom sprint test. While these measurement discrepancies likely contributed to heterogeneity, the established reliability and validity of all incorporated assessment methods partially mitigated methodological influences on outcome interpretation.

Limitations

When interpreting the findings of this study regarding the impact of RS-COD on player ability, several limitations must be acknowledged. Firstly, there is a scarcity of direct comparative studies between RS-OCOD and RS-MCOD, which may introduce bias when synthesizing data from various investigations. Secondly, the exclusion of five studies due to non-compliance with established intervention definitions suggests inconsistencies and a lack of standardization in intervention protocols across studies (see Supplementary file S1). Variations in intervention schemes may also contribute to the heterogeneity of results (Table 2). Additionally, although we examined and cataloged the measurement devices used in the included studies and found that the methods employed were generally consistent, we cannot dismiss the possibility that minor variations in these methods may have influenced the results. Lastly, the presence of publication bias, as indicated by the asymmetry in the funnel plot and confirmed by Egger’s test, suggests that unpublished studies may affect the comprehensiveness of our conclusions. While sensitivity analysis provided some assurance regarding the stability of the data, any analysis relying on a finite dataset is subject to the constraints of sample selection and data integrity. These limitations should be considered when assessing the effectiveness of RS-COD in enhancing player ability. Future research should aim to address these gaps to provide a more robust evidence base for the incorporation of RS-COD into training regimens.

Conclusion

This systematic review and meta-analysis comprehensively assessed the impact of RS-COD training on players’ jumping, sprinting, and COD abilities. The results indicated that RS-COD significantly enhanced these key sports performance indicators, with RS-OCOD can significantly enhance a player’s CMJ ability, while RS-MCOD can notably improve sprinting and COD abilities. These findings support the application of RS-COD as a specialized form of RST in player training to improve competitive performance. Among them, RS-MCOD is prioritized for sports requiring frequent directional changes, whereas RS-OCOD enhances vertical jump performance in explosive sports.

Practical Application

Based on the findings of this study, we recommend implementing sport-specific training protocols to optimize performance adaptations. For sports requiring explosive vertical jumps such as basketball and volleyball, RS-OCOD protocols incorporating 180° directional changes followed by vertical jumps should be prioritized, with a training frequency of 2–3 sessions per week, each comprising 6-8 sprints interspersed with 30-second recovery intervals to enhance lower limb explosive power. Sports involving frequent directional changes like soccer and basketball would benefit from RS-MCOD protocols consisting of 3–4 weekly sessions with 6-8 sprints per session (incorporating 3–5 directional changes at 90°–180° angles), implemented with 4–5 min of inter-set recovery and 20–40 s of inter-repetition rest, supplemented by cone drills or agility ladder exercises to simulate game-specific scenarios and improve rapid deceleration-acceleration capabilities. For racquet sports including tennis and badminton, modified RS-MCOD protocols focusing on smaller-range lateral movements are recommended to enhance sidestepping performance. Standardized movement execution should be strictly enforced during training to minimize injury risk, complemented by dynamic recovery activities (e.g., light jogging/stretching) between sets and post-session cryotherapy to facilitate recovery. The training program should integrate eccentric lower limb exercises (e.g., Nordic hamstring curls) with neuromuscular activation drills (e.g., light signal reaction training), while adopting periodized loading strategies—emphasizing RS-MCOD during preseason (12-week program) and transitioning to maintenance training during competitive seasons (1–2 sessions per week, e.g., 2 sets × 4 sprints). Scientific optimization of training parameters (including directional complexity and recovery duration) combined with standardized monitoring protocols can maximize performance gains, while future research should explore potential synergistic effects with cognitive training interventions.

Supplemental Information

Supplemental Information 1 Supplementary Material

Supplemental Information 2 Data

Supplemental Information 3 PRISMA checklist

Additional Information and Declarations

Competing Interests

Author Contributions

Human Ethics

Data Availability

The authors declare there are no competing interests.

Runzhou Kong conceived and designed the experiments, performed the experiments, analyzed the data, prepared figures and/or tables, and approved the final draft.

Lei Cao analyzed the data, prepared figures and/or tables, and approved the final draft.

Dongyu Li analyzed the data, authored or reviewed drafts of the article, and approved the final draft.

The following information was supplied relating to ethical approvals (i.e., approving body and any reference numbers):

The present systematic review and meta-analysis followed the PRISMA guidelines, and the protocol has been registered in PROSPERO (ID: CRD42024620508).

The following information was supplied regarding data availability:

The raw measurements are available in the Supplementary File.

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
