# Peer review of "The chronic effects of change of direction during repeated-sprint training on jumping, sprinting, and change-of-direction abilities in players: a systematic review and meta-analysis"

_PeerJ, doi:10.7717/peerj.19416_

## Round 0.1 · original submission · Minor Revisions

Dear Author,
I have read your manuscript and all comments of reviewers very carefully. You should read all comments and improve your manuscript.
Kind regards,
Editor

**Language Note:** The review process has identified that the English language must be improved. PeerJ can provide language editing services - please contact us at [email protected] for pricing (be sure to provide your manuscript number and title). Alternatively, you should make your own arrangements to improve the language quality and provide details in your response letter. – PeerJ Staff

·

Basic reporting

I read this manuscript carefully and I must highlight it is written in clear, professional English, suitable for an international audience. The structure conforms to PeerJ's suggested format, with well-defined sections including the introduction, methods, results, discussion, and conclusion. References are appropriately cited, covering relevant previous literature which establishes a sound context for the research question. Figures and tables are relevant, well-labelled, and support the manuscript's findings effectively. All raw data seems to have been shared in accordance with PeerJ’s Data Sharing policy. However, I have some minor recommendation. I would appreciate if the authors consider expanding the introduction to provide a more detailed discussion of the biological mechanisms underpinning the observed effects of RS-COD training. This could offer readers a deeper understanding of the potential physiological implications. In addition, I would also recommend if the authors enhance some figures (specifically Figures 2 and 4) by increasing the resolution and improving the color contrast to make them more accessible for color-blind readers.

Experimental design

With regards to experimental design, I would highlight that the research question is clearly defined, and the manuscript addresses a significant gap in knowledge regarding the specific impacts of RS-COD compared to other training modalities. On the other hand, the methods are robust, employing a systematic review and meta-analysis framework adhering to PRISMA guidelines, which strengthens the reliability of the findings. Furthermore, the study design, including the selection of studies and data analysis, is rigorous and well-documented, ensuring reproducibility. However, I have also to recommend some minor corrections. Namely, the authors should clarify why certain databases were selected for the literature search and whether any relevant databases might have been excluded. In addition, I would also recommend if the authors provide further details on the process of data extraction and analysis, particularly how disagreements were resolved during the screening of studies.

Validity of the findings

With regards to Validity of the findings I would highlight that the findings significantly contribute to the existing body of knowledge by demonstrating the differential effects of various RS-COD protocols on athletic performance. I would also confirm that conclusions are well supported by the data, directly tied to the research question, and confined within the scope of the results. However, this section did pass without some minor recommendations for revision. Namely, although subgroup analyses were performed, the authors should discuss the potential for type I errors due to multiple comparisons and consider adjusting for this statistically. In addition, for the reason the authors mentioned the publication bias in the context of jumping ability, I would also recommend them to discuss the implications of this bias more thoroughly and consider additional strategies to mitigate its impact.

Additional comments

My additional comments are focused on valuable contribution of this manuscript to the literature on sports science and training methodologies as the systematic approach and the meta-analysis conducted are both rigorous and of high quality, as well as to the ethical considerations and data availability statements that are appropriately included, which enhances the transparency and reproducibility of the work. However, future updates of this review could benefit from a broader search strategy to include studies in other languages or unpublished datasets to minimize publication bias. Hence, it would be beneficial for the authors to include a section on practical applications of these findings in athletic training programs.

For the reason, the manuscript is well-prepared, with a few areas for improvement that would enhance clarity and depth, I do believe this manuscript should be well-suited for publication in PeerJ with the recommended minor revisions.

Reviewer 2 ·

Basic reporting

If I look at this work as a systematic review, I think it is well done. I don't see any technical flaws. Everything is shown precisely. On that side, I would recommend that the manuscript be published in its current form.
However, I have no experience in preparing a meta-analysis. I would say that it meets all standards, but I would recommend that the manuscript be looked at by another reviewer who is more expert in meta-analysis than I am.

Experimental design

If I look at this work as a systematic review, I think it is well done. I don't see any technical flaws. Everything is shown precisely.
However, I have no experience in preparing a meta-analysis. I would say that it meets all standards, but I would recommend that the manuscript be looked at by another reviewer who is more expert in meta-analysis than I am.

Validity of the findings

Summarized findings and new conclusions about repeated sprint training will be useful for practitioners. The possibility of applying existing knowledge has been increased, and some of the author's new explanations will be useful.

Additional comments

I would like to emphasize again that the greatest quality of this study is the possibility of applying already existing knowledge, and some new explanations of the author will be useful.

·

Basic reporting

This study appears to be novel, and author showed an interesting point about “The impact of change-of-direction repeated sprint training on players' jumping ability, sprint speed, and change-of-direction ability: A systematic review and meta-analysis”.
The authors have conducted a high-quality meta-analysis with clear objectives and a sound methodological approach.
Additional discussions on variability among studies, methodological differences, and potential limitations would enhance the scientific value of the manuscript.
The manuscript is written in professional and academic language. However, certain sections, particularly the discussion, could be clearer, as some conclusions based on different studies need more precise articulation.
Based on what I have read, I notice a few things that would be good to correct, in order to improve the quality of the article.

Experimental design

A deeper explanation of the mechanisms behind RS-MCOD’s stronger effects compared to RS-OCOD, as well as the practical sports implications of these findings, would be beneficial.
The introduction provides sufficient context and references relevant literature. However, a deeper discussion on the differences between RS-OCOD and RS-MCOD would enhance the study’s value.
The structure follows PRISMA guidelines and adheres to PeerJ journal standards.
The figures (forest plots, funnel plots) are appropriate, but the explanations of the results could be clearer. Additionally, more discussion on potential publication bias in the included studies would be beneficial.
Relevance of the Research Question:
The study addresses an important topic in sports science and fills a gap in the literature regarding the effects of RS-COD on different aspects of athletic performance.
Methodology and Reproducibility:
The methodology is well described, including inclusion/exclusion criteria, database search strategy, and data analysis. However, more details on the specific tests used to measure performance (beyond their simple mention) would be useful.
Write the P value in lowercase if it indicates statistical significance.
Consistently use the number of decimal places throughout the text. In some places there are 2, and in others 3 and 4. It will be clearer to readers when you unify these parameters.

Validity of the findings

The study assesses the risk of bias, but a more detailed discussion of the potential limitations of different studies and their impact on the findings would strengthen the manuscript. Additionally, the study selection criteria could be further elaborated.
The results are well presented through meta-analytic indicators (SMD, I², p-values).
The high heterogeneity in sprint speed (I²=71%) and COD ability (I²=70%) suggests variability among studies, which is not sufficiently addressed in the discussion.
Interpretation and Connection to the Research Question:
The conclusions are supported by the data, but they should be interpreted more cautiously considering publication bias.
The subgroup analysis reveals differences between RS-OCOD and RS-MCOD, but the manuscript does not fully explain why RS-MCOD significantly improves sprint and COD ability while RS-OCOD does not show statistically significant effects.
The limitations are acknowledged, but more discussion is needed on how they can be addressed in future research.
The issue of publication bias is mentioned, but without suggestions on how it might influence the interpretation of the results.
Upon reviewing the manuscript, it is evident that a significant number of references are missing within the provided PDF document. This omission hinders a thorough assessment of the study’s foundation and the credibility of the cited sources. I strongly encourage the authors to ensure that all references are properly included in the final version to allow for a complete evaluation.
Furthermore, the majority of the cited references are more than five years old, with several being significantly older. While foundational studies are valuable, it is essential to incorporate more recent research to reflect the latest advancements in the field. Sports science, particularly in the areas of repeated sprint training and change-of-direction performance, is rapidly evolving, and the inclusion of up-to-date literature (from the last five years) is crucial to maintaining the manuscript’s relevance and scientific rigor.

Additional comments

I urge the authors to conduct a more comprehensive review of the recent literature and integrate studies published within the last five years. This will not only strengthen the theoretical framework and discussion but also enhance the manuscript’s overall impact and reliability.

Reviewer 4 ·

Basic reporting

Dear Authors,
Thank you for submitting your manuscript. The study concept is of practical interest and has strong conceptual merit. However, after careful evaluation, I have significant concerns regarding the execution of the meta-analysis, which limits the robustness of the findings. Unfortunately, in its current form, I cannot recommend the manuscript for further consideration. Below, I outline key methodological issues and provide suggestions for improvement.

Experimental design

Your study includes nested data structures, where the same study contributes multiple effect sizes from different groups or repeated measures within the same group. However, you vaguely stated that effects were aggregated. This approach is statistically inappropriate, as it does not account for the hierarchical dependency of the data. Simply averaging or aggregating effect sizes can lead to inflated precision and erroneous conclusions. The appropriate approach for such datasets is multi-level meta-analysis, which has been well-documented for handling dependent effect sizes (e.g., Cheung, Neuropsychology Review, 2019; Fernández-Castilla et al., Behavior Research Methods, 2020).

Additionally, I recommend providing more detailed information on your methodology, particularly regarding inclusion criteria (e.g., the player characteristics for inclusion are vague, whether study design included cross over design including none, single, or multiple CoDs, or model these as a moderating effect), data extraction procedures, how missing data was handled, and statistical procedures. For guidance on best practices in conducting systematic reviews and meta-analyses in Exercise, Rehabilitation, Sport Medicine, and Sport Science, I suggest referring to the paper by Ardern et al. 2022. Finally, from previous experience of the literature and considering your (general) inclusion criteria: healthy players, ‘implementation of repeat sprint training with change of direction’, no comparators and any ‘assessment of jumping ability’, I believe there is vast of research that could be included in the meta-analysis.

Validity of the findings

As per my previous comment, the current approach to handling multiple effect sizes does not account for the hierarchical dependency of the data, and can significantly impacts the validity of the findings. The use of multi-level meta-analysis is strongly recommended to ensure that the estimates accurately reflect the true variability within and between studies.

Additional comments

No additional comments

·

Basic reporting

Writing - The overall quality of writing is low and needs significant improvements. I recommend that the current investigators collaborate with an additional author who has strong English writing skills.
Sufficient context and background is provided.
Professional structure, figures and tables are provided, although Table 1 requires some improvements.

Experimental design

Fits in with the aims and scope of the journal. The research fills an identified knowledge gap, although this could be more clearly explained within the introduction.
Important details are missing throughout the methods.
The research is performed with conformity to ethical and professional standards.

Validity of the findings

Statistical analysis can be improved inline with more modern approaches in sports science. Overall, much more practical applications of the findings are needed throughout the discussion and conclusions to have impact.

Additional comments

It is an interesting area of investigation which provides some novel insights. However, there are some major area’s that require attention. While considerable improvements are required, if they can be achieved, then the paper would be beneficial for practitioners.

---

## Round 0.2 · accepted · Accept

Dear author,
I have read your improved manuscript and the reviewer's comments very carefully and in my opinion the manuscript is of good quality at this point and ready for publication in PeerJ.

·

Basic reporting

I have no further requirements.

Experimental design

I have no further requirements.

Validity of the findings

I have no further requirements.

Additional comments

No additional comments.